# Sucralose Targets the Insulin Signaling Pathway in the SH-SY5Y Neuroblastoma Cell Line

**DOI:** 10.3390/metabo13070817

**Published:** 2023-07-04

**Authors:** Marina Čović, Milorad Zjalić, Lovro Mihajlović, Marianna Pap, Jasenka Wagner, Dario Mandić, Željko Debeljak, Marija Heffer

**Affiliations:** 1Department of Medical Biology and Genetics, Faculty of Medicine, Josip Juraj Strossmayer University of Osijek, 31000 Osijek, Croatia; 2Department of Pharmacology and Biochemistry, Faculty of Dental Medicine and Health, Josip Juraj Strossmayer University of Osijek, 31000 Osijek, Croatia; 3Department of Molecular Medicine and Biotechnology, Faculty of Medicine, University of Rijeka, 51000 Rijeka, Croatia; 4Department of Medical Biology and Central Electron Microscopic Laboratory, University of Pécs Medical School, 7624 Pécs, Hungary; 5Clinical Institute of Laboratory Diagnostics, Osijek University Hospital, 31000 Osijek, Croatia; 6Department of Medical Chemistry, Biochemistry and Clinical Chemistry, Faculty of Medicine, Josip Juraj Strossmayer University of Osijek, 31000 Osijek, Croatia; 7Department of Pharmacology, Faculty of Medicine, Josip Juraj Strossmayer University of Osijek, 31000 Osijek, Croatia

**Keywords:** non-nutritive sweetener, insulin signaling, dopaminergic neurons, insulin like growth factor 1 receptor, neurodegeneration

## Abstract

Sucralose is widely used as a non-nutritive sweetener (NNS). However, in order to justify its use as a non-nutritive food additive, sucralose would have to be metabolically neutral. The aim of this study was to examine whether sucralose altered the insulin signaling pathway in an in vitro cell model of Parkinson’s disease (PD)—the dopaminergic differentiated cell line SH-SY5Y. Cells were exposed to sucralose alone and in combination with either insulin or levodopa. Activation of the insulin signaling pathway was assessed by quantifying protein kinase B (AKT) and glycogen synthase kinase 3 (GSK3), as well as the phosphorylated forms of insulin-like growth factor 1 receptor (IGF1-R). Metabolic effects were assayed using MALDI-TOF MS analysis. In the cell viability test, 2 mM sucralose had a negative effect, and levodopa in all combinations had a positive effect. Sucralose treatment alone suppressed GSK3 and IGF1-R phosphorylation in a dose-dependent manner. This treatment also altered the metabolism of fatty acids and amino acids, especially when combined with insulin and levodopa. Suppression of the insulin signaling pathway and sucralose-induced changes in the metabolic profile could underlie a diet-acquired insulin resistance, previously associated with neurodegeneration, or may be an altered response to insulin or levodopa medical therapy.

## 1. Introduction

Non-nutritive sweeteners (NNSs) are functional additives that impart sweetness to food and beverages, while having negligible nutritive (caloric) value [1]. The food industry widely employs them as alternatives to added sugars, which are being linked to the development of various chronic diseases [2]. The scientific findings about the harmfulness of extra calories intake and the human craving for sweets have jointly contributed to the diversity of NNSs and the processed/ultra-processed foods (or even food supplements) to which they are added. However, instead of contributing to reducing the obesity pandemic, the microbiome and many other studies showed sweeteners may further exacerbate it [3,4]. Therefore, recently published guidelines by the World Health Organization (WHO) [5] recommend against artificial sweeteners use to control body weight or reduce the risk of noncommunicable diseases. The recommendations were made mostly on the basis of a meta-study that included 283 individual studies with the conclusion that acute use of NNSs does not pose a health risk, but there is not enough evidence to conclude the same about chronic use [5]. Before further scientific research inspires more careful use of sweeteners, and sugars in general, in the food industry, we should identify NNS species and populations that are at risk due to its consumption, given the thus far recognized potential pathophysiological effects.

Although the entire group of artificial sweeteners acts primarily through the sweet receptors, they are chemically different molecules that have a different affinity for the receptor and a different metabolic fate, so the effects of one cannot be extrapolated to the others. For this reason, in this study, we deal with sucralose, a chlorinated derivative of sucrose that is widely used and is known as one of the most potent (600 times sweeter than sucrose). The perception of sweetness on the tongue is physiologically related to the preparation of the organism for digestion and storage of the appropriate amount of sugar. Not only the tongue, but a variety of cells, including the beta cells of the pancreas [6] and the satiety nuclei in the hypothalamus [7], express sweet taste receptors, including functional signaling pathways that directly lead to the secretion of insulin in the first case or serve as a sensor for the amount of glucose in the blood in the second case.

In addition to regulating the secretion of insulin, sweet receptors also affect the secretion of other hormones involved in glycemic control. For example, using the sweet taste receptor coupled to gustducin, glucose given orally stimulates the endocrine L cells to secrete the Glucagon-like peptide-1 (GLP-1) [8] hormone that stimulates insulin release and survival of beta cells, inhibits glucagon release, slows gastric emptying and increases satiety [9]. Although sucralose would be expected to increase the antidiabetogenic effects of GLP-1, this does not necessarily occur. A study on obese participants with controlled glycemia [10] and a following study of participants with normal weight [11] showed elevated glucose (in all participants) and insulin levels (in obese participants) on the oral glucose tolerance test after sucralose ingestion. Contrary to acute administration, chronic administration of sucralose in young healthy adult subjects in a double-blind controlled study led to significant hyperinsulinemia [12]. Hyperinsulinemia is an early sign of metabolic dysfunction [13] and a potential pathophysiological mechanism of the obesogenic action of sucralose. Another obesogenic mechanism is a direct effect on hypothalamic satiety centers and appetite regulation [14]. Independent of obesogenic effects, in a recent prospective clinical trial, the total intake of NNS was associated with the risk of cardiovascular disease [15], which speaks in favor of the association between endothelial dysfunction and sweetener use. The case is by no means closed because there are systematic reviews and meta-analyses that say exactly the opposite—the use of NNS is not obesogenic and does not pose a cardiovascular risk [16,17,18].

In this study, we assume a diabetogenic effect of sucralose, based on interaction between two signaling pathways: the sweet taste receptor and the insulin receptor. This interaction is particularly harmful to cells whose metabolism depends on glucose, such as neurons, because the activation of the insulin pathway in a state of glucose deprivation puts the cells under stress. Dopaminergic neurons are a good example of vulnerable cells whose survival involves a gastrointestinal cue–reward association [19]. Enteric dopaminergic neurons could be particularly sensitive [20], while the effect of sucralose on central dopaminergic neurons depends on its ability to cross the blood–brain barrier. 

Two therapies have a potential neuroprotective effect: insulin [21] and levodopa [22]. The positive effects of insulin consist of stimulating the survival and synaptic plasticity of neurons, regulating glucose metabolism and the release of neurotransmitters, and maintaining the integrity of the blood–brain barrier. A chronic imbalance of insulin metabolism, which manifests as type 2 diabetes, is a potential risk for the most common non-heritable neurodegenerative diseases [23,24]. Interventions with antidiabetics in terms of preventing or delaying the onset of neurodegeneration symptoms are the subject of numerous clinical studies [25], including intranasal insulin administration [26], currently without cognitive benefits. Levodopa is a precursor of dopamine and a symptomatic therapy for PD. A toxic metabolite called 3,4-dihydroxyphenylacetaldehyde (DOPAL), which is produced from dopamine by the action of the enzyme monoamine oxidase B (MAO-B) [27], reduces the positive effects of levodopa therapy. Despite this, the levodopa/carbidopa combination is the most commonly prescribed medication for PD. 

In this study, we use the human SH-SY5Y cell line, which expresses sweet taste receptors and takes on a retinoic acid-induced dopaminergic phenotype upon differentiation [28,29], hence serving for this reason as the cell culture model for Parkinson’s disease (PD). Upon exposure to sucralose, cells were treated with insulin to test insulin signaling or interference with levodopa as a standard therapy for PD.

## 2. Materials and Methods

### 2.1. Cell Culture

The human neuroblastoma cell line SH-SY5Y, ATCC CRL-2266 (Manassas, VA, USA), was used. The cells were grown in DMEM/F12 high glucose (4.5 g/L) medium (Capricorn Scientific, Ebsdorfergrund, Germany) supplemented with 3 mM L-glutamine, 0.1 mg/mL streptomycin, 100 IU penicillin G (L-Glutamine-Penicillin-Streptomycin solution (Sigma-Aldrich, Saint Louis, MO, USA)), 1x non-essential amino acids (Sigma-Aldrich, Saint Louis, MO, USA), 15 mM HEPES (Sigma-Aldrich, Saint Louis, MO, USA), 10% fetal bovine serum (FBS) (Capricorn Scientific, Ebsdorfergrund, Germany) and 1 mM sodium pyruvate solution (Capricorn Scientific, Ebsdorfergrund, Germany). Cells were differentiated using 10 mM all-trans-retinoic acid (Merck, Kenilworth, NJ, USA) for 9 days (Figure 1). For each experiment, cells were grown and differentiated in at least three biological replicates. Each experimental group had a corresponding control group without treatment with sucralose (SUC) (Esarom, Oberrohrbach, Austria) and insulin (Thermo Fisher Scientific, Waltham, MA, USA) or levodopa (Merck, Darmstadt, Germany). For protein analysis by Western blot and metabolome analysis using matrix-assisted laser desorption/ionization time-of-flight mass spectrometry (MALDI-TOF MS), cells were grown, differentiated and treated in 6-well plates (TPP Techno Plastic Products AG, Trasadingen, Switzerland), and for the MTT viability test, in 96-well plates (Corning Incorporated, Corning, NY, USA).

### 2.2. Treatments

Differentiated cells were separately treated with 3 concentrations of SUC (0.2 mM, 2 mM, 20 mM) for 24 h, in DMEM/F12 high glucose (4.5 g/L) medium without FBS and antibiotics. Treatment was fixed to 24 h based on a previous pharmacokinetic study [30]. Three concentrations of sucralose were used to determine the dose response. Afterwards, the cells were treated with either insulin (2 μg/mL) or levodopa (1 μg/mL), for 1 h. Control cultures had no SUC pre-treatment. The cells were then homogenized with an ultrasonic homogenizer (Bandelin Sonopuls 2070, BANDELIN electronic GmbH & Co. KG, Berlin, Germany) in 1x phosphate buffered saline (PBS) with protease inhibitors (Figure 1 and Table 1). The middle value of the 3 concentrations of SUC used corresponds to the permissible daily intake for a person weighing 60 kg [31,32]. The 10-times-lower concentration (0.2 mM) was motivated by a study that detected NNS in the urine of individuals, who were exposed due to widespread sucralose in their daily diets [33]. A 10-times-higher concentration (20 mM) was chosen to exceed twice the permitted daily intake. This concentration (20 mM) was used to assess the neurotoxicity of SUC.

### 2.3. MTT Assay

The cytotoxic effect was determined by the MTT viability assay, according to the manufacturer’s instructions (Abcam, Cambridge, UK). MTT estimates the number of living cells based on the activity of a mitochondrial enzyme that converts the yellow MTT salt [3-(4,5-dimethylthiazolyl-2)-2,5-diphenyltetrazolium bromide] into a purple formazan product. MTT was added to wells with the treated cells and incubated for 2 h at 37 °C. Formazan crystals were dissolved using 100 µL of MTT solubilization solution per well (Table 1). The color intensity of the solution was then read at 595 nm using a microplate absorbance reader (iMarkTM, Bio Rad, Hercules, CA, USA). 

### 2.4. Protein Assay

The total amount of proteins in the samples was measured by the Bradford method in flat-bottomed microtiter plates, according to the manufacturer’s instructions (BioRad, Hercules, CA, USA) [34]. A total of 1 μL of triplicate samples and 200 μL of Bradford reagent (Thermo Scientific Pierce Coomassie Plus Protein Assay, Waltham, MA, USA) were added to the wells. The absorbance was read on the microplate absorbance reader (iMarkTM, Bio Rad, Hercules, CA, USA) at 595 nm.

### 2.5. SDS-PAGE Electrophoresis and Western Blot

Protein analysis was performed by SDS-PAGE electrophoresis and the Western blot method. Protein denaturation was performed by boiling samples in the Laemmli buffer (Table 1) at 100 °C for 5 min. Self-made 12% acrylamide gels (Acros Organics, Morris Plains, NJ, USA) with the addition of 1.25% of 2,2,2-trichloroethanol (Acros Organics, Morris Plains, NJ, USA) for stain-free load normalization and 5% stacking gels were used. A total of 3 μL of protein standard (SeeBlue 2 Plus, Thermo Fisher Scientific, Waltham, MA, USA) and 15 μg of proteins of each sample were loaded to gel. Electrophoresis was performed in a Hoefer SE250 Mighty small II mini vertical unit system (Hoefer Inc., San Francisco, CA, USA) with an electric field strength of 15 mA per gel in electrophoresis buffer (Table 1) with constant cooling. The Western blot method was performed in a Hoefer TE22 Mighty Small transfer tank system (Hoefer Inc., San Francisco, CA, USA) with an electric field strength of 200 mA in Towbin buffer (Table 1) for 2 h with constant cooling. Polyvinylidene difluoride membranes (PVDFs) (ThermoFisher Scientific, Waltham, MA, USA) were incubated in 3% BSA blocking solution (Table 1) for 30 min and then incubated with highly specific primary antibodies (Table 2) in a ratio of 1:1000 for up to 24 h at 4 °C with continuous rotation. After incubation, the membranes were washed in PBS-T buffer: PBS buffer (Table 1) with the addition of Tween-20 detergent (Fisher Scientific, Branchburg, NJ, USA). Then, the membranes were incubated in a 3% BSA solution with the appropriate secondary antibody at room temperature (Table 2) for 2 h with continuous rotation. The membranes were washed, incubated in the Immun-Star™ WesternC™ Chemiluminescence Kit according to the manufacturer’s protocol (BioRad, Hercules, CA, USA) and visualized with a visualization system (ChemiDocTM Imaging system, BioRad, Hercules, CA, USA). The obtained results were analyzed and quantified using the Image J program (NIH, Bethesda, MD, USA).

### 2.6. MALDI-TOF MS

Metabolomics of the treated differentiated SH-SY5Y was performed by MALDI-TOF MS on a Bruker UltrafleXtreme (Bruker Daltonik GmbH, Bremen, Germany). Cell homogenate was prepared in 150 mM ammonium acetate buffer pH 8.2 with all inhibitors used in the protocol for Western blot analysis. Everything was performed in three biological replicates. Proteins in homogenate were measured with the previously described Bradford reagent and protocol. Homogenate was diluted to an amount of 2.5 μg/μL of protein concentration. A total of 160 μL of homogenate was pipetted into a 1.5 mL Eppendorf tube, and 400 μL of HPLC-grade methanol (J. T. Baker, Phillipsburg, NJ, USA) was added and vortexed. A total of 200 μL of chloroform of analytical purity (Carlo Erba, Milan, Italy) was added and vortexed; this amounts to a ratio of 0.8:2:1 for sample: methanol: chloroform. The mixture was left in ice for 10 min to complete lipid extraction. After 10 min, 200 μL of chloroform and 200 μL of HPLC-grade water (Carlo Erba, Milan, Italy) was pipetted. The resulting solution was vortexed for 10 s and centrifuged at 20,000 g for 5 min at 4 °C. After centrifugation, three phases were visible: upper polar water phase, middle protein phase and bottom non-polar chloroform/methanol phase. With a pipette, the top phase was removed to an amber glass HPLC collection vial, followed by collecting the bottom phase for transfer into a separate amber glass HPLC collection vial [35]. 

Solvents were removed in a stream of nitrogen while heated up to 45 °C, using Techne Sample Concentrator with a DriBlock heater (Bibby Scientific UK, Beacon Road, Stone, Staffordshire, UK). After, solvent evaporation samples were dissolved in 100 μL of methanol:chloroform in a ratio of 2:1. Two matrices were used: dihydroxybenzoic acid (Sigma-Aldrich, St. Louis, MO, USA) for positive mode and 9-aminoacridine (Merck, Darmstadt, Germany) for negative mode. Both matrices were prepared in a concentration of 10 mg/mL and dissolved in HPLC-grade methanol (J. T. Baker, Phillipsburg, NJ, USA). Matrix and sample were mixed in a 1:1 ratio total equaling 6 μL for each sample. A total of 2 μL of the matrix/sample mixture was applied on a polished steel plate sample carrier and quickly dried with forced air current at room temperature. At one spot, red phosphorus (Sigma-Aldrich, St. Louis, MO, SAD) was applied to be used for negative and positive mode calibration. Samples were analyzed in the positive and negative mode in the range 200–1600 *m*/*z* in 2 steps. The first step range was 200–1000 *m*/*z*, and the second step range was 1000–1600 *m*/*z*. For each step, two technical replicas of every sample isolate were applied onto a MTP 384 target plate polished steel (Bruker Daltonik GmbH, Bremen, Germany). The MALDI-TOF settings of the Bruker Ultrafle Xtreme were identical for the positive and negative mode. The sample rate and digitizer were set at 5.00 GS/s, the Smartbeam (laser) diameter was set to medium, laser shot frequency 2000 Hz, laser power 90%, 200 shots/pixel, and every measurement was a sum of 15,000 shots. The results of analysis and data extraction were performed in Bruker FlexAnalysis v3.4 (Build 79) software (Bruker Daltonik GmbH, Bremen, Germany) with the following settings: peak detection algorithm was Snap; s/n threshold was 6, smoothing algorithm was Savitzky–Golay (width 0.2 *m*/*z*, cycles 1), Baseline subtraction was conducted with TopHat. Mass-to-charge ratio (*m*/*z*) signals and their intensities were statistically processed in the R software environment v4.0.3. (R Foundation for Statistical Computing, Vienna, Austria) using the Kyoto Encyclopedia of Genes and Genomes (KEGG) and FELLA packages.

### 2.7. Statistics

Significance levels for all statistical tests were set at α = 0.05. Obtained data were tested for normality of distribution with the Shapiro–Wilk test. The homogeneity of variances was tested with Bartlett’s F-test. The samples were analyzed for two variables: concentration of sweeteners and treatment with insulin and levodopa. In the case of normal distribution and homogeneous variance, two-way ANOVA was used. In the case of statistically significant results of two-way ANOVA, statistically significant differences between groups were tested by the Tukey HSD post hoc test. All *p*-values less than 0.05 were considered statistically significant. In the case of two-way ANOVA, data are presented as the mean value with the associated standard deviation. The Welch *t*-test was used to analyze the data obtained by MALDI-TOF MS imaging. Differences in the signal intensities of each detected compound were tested separately. The level of significance was set at α = 0.05 with a false discovery rate (FDR) correction of the *p*-value, as false positive results would be avoided. R software v4.0.3 (R Foundation for Statistical Computing, Vienna, Austria) was used for all statistical analysis. Data were presented as a compact letter display graph. The bars correspond to an average value of the calculated percentage with a confidence interval of ±95%. 

## 3. Results

### 3.1. Sucralose Combined with Levodopa Increased Cell Viability

To determine whether sucralose, sucralose + insulin, or sucralose + levodopa impacts cell viability, an MTT assay was performed. The assay is based on a reagent metabolized by mitochondria that accumulates only in metabolically preserved cells, allowing the observed metabolic changes to be attributed to treatment rather than cell death.

Sucralose itself, apart from the 2 mM concentration, did not exert a negative effect on cell viability (Figure 2). Nevertheless, when 2 mM sucralose was applied, a small but significant decrease in viability was observed. From this, we concluded that sucralose applied at a 2 mM concentration interferes with mitochondrial function. In contrast, insulin treatment did not affect cell viability.

Levodopa had a singular positive effect on cell viability. Applied alone, levodopa increased cell viability by 24% compared to untreated controls. When sucralose treatment was paired with levodopa, a 44% and 37% increase in cell viability was observed for 0.2 mM and 20 mM sucralose, respectively. Even when treated with 2 mM sucralose, levodopa was shown to have a protective effect, which supports its antioxidant effects [22].

The results of the MTT assay show that selected concentrations of sucralose are not toxic as such, while levodopa has the expected positive effect on the survival of cultured dopaminergic neurons, probably due to the alleviation of oxidative stress or the activation of the dopamine survival signaling pathway.

### 3.2. Suppresion of Insulin Signaling Pathway after Sucralose Treatment

The first objective of the study was to determine the effect of sucralose on the insulin signaling pathway. AKT is a key molecule in cellular metabolism that controls several downstream signaling pathways. Relevant to this study, activation of PI3K (Phosphoinositide 3-Kinase) in the insulin signaling pathway leads to activation of the mammalian target of rapamycin complex 2 (mTORC2), which in turn phosphorylates AKT at Serine 473 (Ser473) and promotes its full activation [36]. The maximum phosphorylation of both AKT and PI3K is expected to be reached about an hour after insulin stimulation [37]. Since pAKT serves as a cellular hub and reflects well the activation of the insulin signaling pathway, we focused on it first.

Sucralose alone and additional treatments with insulin and levodopa did not induce significant changes in AKT levels (Figure 3A). As expected, phosphorylation of AKT (pAKT) in cells not treated with sucralose occurred after insulin treatment (Figure 3B). Interestingly, treatment with levodopa also caused an increase in pAKT, probably due to the activation of the dopamine signaling pathway. Addition of sucralose had the opposite effect on pAKT, with the response to levodopa attenuated at the 0.2 and 2 mM sucralose concentrations, and the response to insulin at the 2 mM sucralose concentration.

One significant effect of pAkt is its inactivation by the phosphorylation of glycogen synthase kinase-3 (GSK3). GSK3 inhibits glycogen synthase, thus reducing glycogen synthesis and promoting its breakdown. Unlike some other neurons, the SH-SY5Y neuroblastoma cell line can store glycogen and expresses two GSK3 isoforms—GSK3α and GSK3β [38]. Phosphorylation at serine 9 (pGSK3α) turns off the enzymatic activity of GSK3α, and phosphorylation at serine 21 (pGSK3β) turns off GSK3β, which enables glycogen synthesis and has many other consequences for cell survival. Therefore, we next checked whether the administration of sucralose affects GSK3 phosphorylation.

In cells not treated with sucralose, but treated with insulin and levodopa, which had higher levels of pAKT, phosphorylation of none of the GSK3 isoforms increased, but phosphorylation of non-phosphorylated GSK3β increased (Figure 3C–G). This would mean the mechanism that leads to glycogen accumulation is self-limiting, and the prolonged action of insulin/levodopa (1 h exposure) acts primarily through the GSK3β/pGSK3β expression ratio. The addition of sucralose, especially at concentrations of 0.2 and 2 mM, significantly lowers both pGSK3 isoforms, further shifting the balance away from glycogen accumulation or leading to its breakdown. Since insulin and levodopa had only a minor effect on pGSK3β after 1 h of treatment, the observed phenomenon is the primary result of sucralose treatment (Figure 2F). In conclusion, sucralose affects the executive molecules of the insulin signaling pathway, which may be reflected in the glucose storage capacity of the cell.

### 3.3. Insulin-Like Growth Factor Receptor 1 Beta Subunit (IGF1-Rβ) Is Downregulated following Sucralose Treatment

Considering the observed effects of sucralose on GSK3α and GSK3β and their role in survival and stress response, we were interested to see if other mechanisms involved in cell survival were also affected. In the SH-SY5Y cell line, a signaling pathway that was shown to be particularly protective against beta-amyloid-induced apoptosis acts via insulin-like growth factors 1 and 2. Downregulation of their receptor, the insulin-like growth factor 1 receptor, reduces scavenging ROS via the PI3K/Akt-Nrf2 signaling pathway [39], contributing to amyloid precipitation. Therefore, the second aim of our study was to determine the expression of IGF1-Rβ after sucralose treatment.

Indeed, sucralose downregulated IGF1-Rβ, and the effect was concentration dependent, with a maximal decrease occurring at the 2 mM concentration (Figure 4). In addition, insulin and levodopa caused a similar decrease in receptor levels compared to the untreated control group. This effect of insulin and levodopa disappeared with the 0.2 and 2.0 mM sucralose treatments and unexpectedly turned into its opposite—enhanced IGF1-Rβ expression—at the 20.0 mM concentration. This unexpected result could be related to the increased cell viability, as seen in the results of the MTT test.

### 3.4. Sucralose Altered the Metabolic Response to Insulin and Levodopa–MALDI-TOF MS Analysis

The third objective of our study was to determine how characteristic sucralose dose-dependent metabolic changes were altered by an additional challenge with either insulin or levodopa. The metabolic profile of cell lines is expected to be passage-dependent [40]. Therefore, cells of the same passage were used for all experiments. In order to obtain a more uniform metabolic profile of the cells, SH-SY5Y neuroblastomas were subjected to differentiation. In this way, we tried to avoid changes typical of different phases of the cell cycle or stages of differentiation in order to enhance treatment-specific changes. During KEGG database searches, only unique hits were retained, while multiple hits, including isobars, were excluded, which further reduced the number of identified compounds. Finally, an untreated group was compared with 11 differently treated groups (insulin, levodopa and 3 concentrations of sucralose).

Among the 11 treatment groups, only 9 putatively identified compounds changed significantly compared to the untreated control group—1 sphingolipid metabolite, 6 fatty acids and 2 amino acid derivatives (Table 3). In cells not treated with sucralose, there was no change in the metabolic profile after one hour of insulin treatment, probably due to the length of treatment that reached the equilibrium point. Treatment of control cells with levodopa for one hour lowered the amount of long-chain saturated fatty acid–docosanoic acid. This is a very interesting finding, since the lengthened saturation of fatty acid correlates with the accumulation of α-synuclein in neuroblastoma cells [41]. Namely, Lewy bodies are composed of not only protein precipitates, but also membrane material, so reducing the fatty acid load could be beneficial for dopaminergic neurons.

When the cells were treated with only different concentrations of sucralose, the response to the highest concentration applied was unexpectedly completely absent. Treatment with a low dose of sucralose decreased arachidic acid, another saturated long-chain fatty acid, and led to the appearance of 4-hydroxy-phenyl-acetyl-glutamic acid (4-HPAG). Although the effect on long-chain fatty acids could be interpreted in the same way as the effect of levodopa, the appearance of 4-HPAG, an intermediate compound in the metabolism of phenylalanine and tyrosine, is characteristic of certain metabolic disorders, such as phenylketonuria, alkaptonuria or tyrosinemia [42]. This could be particularly important for children with hereditary metabolic diseases who are advised to avoid aspartame, but not other sweeteners, and who may otherwise be exposed to higher concentrations of NNS due to their dietary limitations [43]. Sucralose applied at a 2 mM concentration led to a decrease in palmitoleic acid, a monounsaturated omega-7 fatty acid. Palmitoleic acid belongs to the lipokines–lipid hormones with anti-inflammatory effects that are involved in communication between the liver and adipose tissue and regulate lipid metabolism [44]. Its role in dopaminergic cells is unknown.

### 3.5. Insulin Plus Sucralose Increases Long-Chain Fatty Acid, Sphingolipid and Tryptophan Metabolites

Although insulin alone had no effect on control cells, administration in combination with sucralose exposure had metabolic consequences. Sucralose applied at concentrations of 0.2 and 2 mM led to a twofold increase in eicosenoic acid, a monounsaturated omega-9 fatty acid. Sucralose at a concentration of 2 mM also led to an increase in arachidic acid, which has already been mentioned in the context of levodopa. The influence of the insulin pathway on lipid synthesis is not unexpected, as lipids also serve as modulators of insulin sensitivity [45]. Insulin administered at the highest concentration of sucralose (20 mM) led to the appearance of 4-hydroxysphinganine and a fivefold increase in 6-hydroxymelatonin. 4-hydroxysphinganine is a sphingolipid metabolite. Sphingolipids are not only important molecules of the cell membrane, but also signaling molecules, but there is no data on the biological significance of this particular metabolite. The finding of 6-hydroxymelatonin is somewhat unexpected, but we assume that it should be commented on in the context of tryptophan metabolism, which is associated with the onset of insulin resistance [46].

### 3.6. Administration of Levodopa with High Sucrose Suppresses Lipid Synthesis

Levodopa has different metabolic effects at different concentrations of sucralose. In the presence of the lowest concentration (0.2 mM), docosanoic acid no longer decreased, but myristic acid appeared. Of all its biological roles, it makes the most sense in this context to comment on it as a bioactive signaling molecule involved in the synthesis of anti-inflammatory prostaglandins [47]. Levodopa at a 2 mM sucralose concentration led to a twofold increase in docosanoic and arachidic acid and a fivefold decrease in palmitoleic acid. The increase in the two saturated long-chain fatty acids may indicate levodopa at these sucralose concentrations no longer exerts the expected protective effect on dopaminergic neurons. At even higher sucralose concentrations (20 mM), there was a further increase in arachidic acid, but also a decrease in docosanoic acid below the levels detected in control cells; a kind of metabolic reversal in lipid synthesis occurred.

In conclusion, sucralose changed the metabolic profile of cells, as well as their response to insulin and dopamine, particularly reflected in the synthesis of saturated and mono-unsaturated fatty acids.

## 4. Discussion

Sucralose is considered safe for use, which is reflected in the high permitted daily doses (5 mg/kg), but it has a long clearance and acts on signaling pathways that regulate cell metabolism. These make it a potential metabolic disruptor for cells expressing sweet taste receptors, such as dopaminergic neurons. Further, sucralose can interfere with the most common therapy for dopaminergic insufficiency—levodopa.

In this study, we demonstrated the possible influence of sucralose on the insulin signaling pathway and interference with the effects of levodopa on the differentiated human SH-SY5Y cell line. Sucralose applied at a concentration of 2 mM, which is the closest to the permitted intake threshold, reduced cell viability the most and caused the greatest downregulation of survival-associated IGF1-R. Sucralose lowered the phosphorylation of the executive molecules of the insulin signaling pathway, AKT and both GSK3 isoforms, and interfered with the physiological response to insulin and levodopa. The effect on pGSK3 was seen at all tested concentrations, while the effect on pAKT was greater at two lower concentrations. Further, sucralose changed the metabolic response of the cell, especially after additional treatment with insulin and levodopa, and the response was dominated by lipids with potential lipokine action.

### 4.1. Sucralose May Be an Insulin Disruptor

After adsorption in the gastrointestinal tract, the peripheral tissues, including enteric neurons, are the first to be exposed to the sweetener. Nevertheless, the finding of the modulatory effect of sucralose on the sweet taste receptors of the hypothalamus and the regulation of appetite [48] justifies the suspicion that neurons behind the blood–brain barrier may also be exposed. A cell line of hypothalamic neurons (mHypoE-N43/5) was used in the study that proved the association of NNS exposure with ER stress and abnormal axon growth [49] and further clarified the molecular mechanism by which changes in feeding behavior related to chronic exposure to sucralose occur in BALB/c mice [14]. In addition to hypothalamic neurons involved in the regulation of satiety, sweet taste receptors are also expressed on dopaminergic neurons that regulate reward [50]. In fact, dopaminergic neurons are crucial for behavioral conditioning, by which metabolic signals are translated into behavior [51]. Further, dopaminergic neurons are particularly susceptible to neurodegeneration associated with glucose metabolism impairment [52], which manifests clinically as Parkinson’s disease (PD). PD is a multi-system and multi-factorial disease, but most of the risk comes from environmental factors (heritability is estimated at 30%) [53]. A recent study supported by the USA Parkinson’s Disease Foundation revealed that the incidence of PD has increased 1.5 times in the last decade [54]. This is an additional motivation to search for possible environmental factors that contribute to the development of the disease [55]. Pesticides, insecticides and industrial pollution stand out from the many assumed so far, but given that the disease is related to the health of the gastrointestinal tract [56,57] (gut-first hypothesis), additives in ultra-processed food come into consideration. Due to their effect on glucose metabolism, NNSs may also be considered an environmental risk for PD.

The SH-SY5Y line, due to its dopaminergic phenotype, is often used as a cell model for Parkinson’s disease [58] suitable for testing environmental toxins [59,60] and pathophysiological mechanisms of neurodegeneration. In the same model, the link between the development of insulin resistance, α-synuclein expression and mitochondrial dysfunction was established [61]. In fact, the disorder of insulin signaling precedes and possibly triggers further pathology that leads to neurodegeneration. In our study, the interference of sucralose with insulin signaling was monitored using the phosphorylation of kinases AKT and GSK3. Instead of the expected increase in phosphorylation or lack of effect predicted by previous literature [30], there was a decrease, especially of pGSK3. The greatest reduction was caused by a 2 mM sucralose concentration. The additional treatment with insulin and levodopa lowered pGSK3 even more, followed by downregulation of IGF1-Rβ and increased synthesis of arachidic, eicosenoic and docosanoic acid. According to previous literature [62], lowering of pAKT levels is associated with, or even caused by, downregulation of IGF1 signaling. Excessive stimulation of sweet taste receptors could theoretically lead to the activation of negative feedback loops associated with insulin signaling.

### 4.2. Sucralose Alters IGF1 Sensitivity

Suppression of the insulin signaling pathway can have various consequences, including the lowering of the response to insulin, characteristic of the onset of insulin resistance [63] and observed in this study. Among other consequences, dysregulation of lipid metabolism is clinically relevant, but also shown in the metabolic profile of the treated cells. Next, the metabolism of neurotransmitters and synaptic functions are affected [64] and here documented as a changed response to levodopa. Insulin binding to IGF1-R leads to receptors’ autophosphorylation and downregulation [65], as seen in the results on control cells. However, sucralose alone leads to a decrease in IGF1-Rβ. Paradoxically, if cells treated with the highest dose of sucralose are additionally exposed to treatment with insulin or levodopa, the opposite effect occurs—the expression of IGF1-R increases, which is reflected in the greater cell survival visible in the MMT assay. SH-SY5Y is a tumor cell line, and this would potentially be an escape mechanism promoting tumor growth. We can conclude that sucralose affects the insulin signaling pathway and the expression of IGF1-R and potentially impairs their established neuroprotective role [66,67]. To our knowledge, there is no other study dealing with the effects of sucralose on the insulin signaling pathway in SH-SY5Y cells, or the combined administration of levodopa and sucralose.

All cells sense the level of extracellular glucose. Glucose sensing is associated with the regulation of its own metabolism, proliferation, differentiation or survival system [68]—the sensitivity of these molecules is of paramount importance. Molecules that interact with glucose sensors can interfere with learned responses and, in chronic administration, lead to pathological metabolic changes [69]. In the obesity pandemic, the emergence of insulin resistance due to the suppression of the insulin pathway is less expected and, therefore, neglected as a possible basis for diabetes and neurodegenerative diseases. In this context, sucralose and other NNSs certainly deserve additional research [70].

The limitation of our study is the monitoring of a relatively small number of epitopes related to the insulin signaling pathway and only one NNS. In the future study, it should be tested whether changes in insulin signaling promote mechanisms of neurodegeneration. The biological role of the lipids identified in this study should certainly be investigated in detail. The ability of sucralose to cross the blood–brain barrier under various conditions needs to be determined in animal models and human subjects to determine the consequences of sucralose’s action on various populations of dopaminergic neurons.

## 5. Conclusions

In conclusion, sucralose interferes with the metabolism of the differentiated SH-SY5Y cell line. Its metabolic response to sucralose is dominated by changes in lipid composition, with associated impacts on the insulin signaling pathway.

## Figures and Tables

**Figure 1 metabolites-13-00817-f001:**
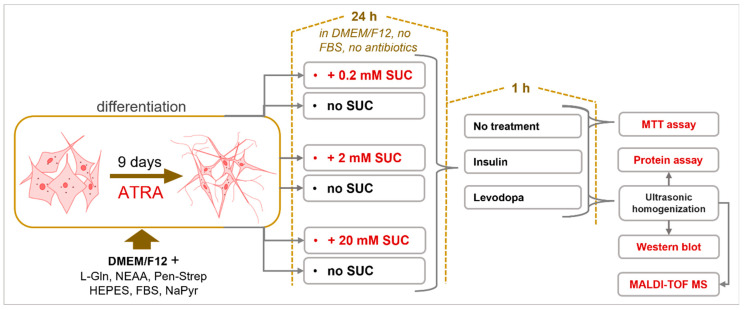
Sucralose Metabolic Impacts on Differentiated Human Neuroblastoma Cells (SH-SY5Y cell line). Cells were differentiated using all-trans-retinoic acid (ATRA) for nine days. Differentiated cells were then treated with three different concentrations of sucralose (SUC). Afterwards, cells were treated with insulin and/or levodopa.

**Figure 2 metabolites-13-00817-f002:**
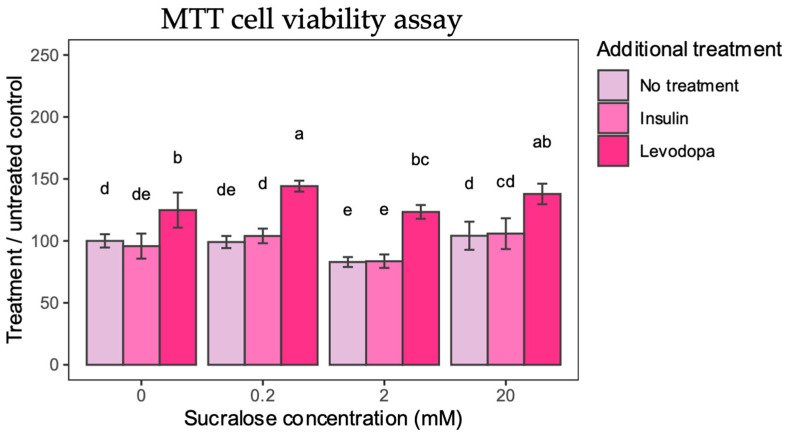
MTT Cell Viability. Cell viability is calculated from absorbance and presented as percentage compared to 0 mM sucralose/No additional treatment group. Two-way ANOVA sucralose-treated groups, F(6,96) = 2.4378, *p* = 0.0308; post HOC test Tukey HSD with *p*-value set at *p* < 0.05. Groups bearing the same letter above the graph are not statistically different.

**Figure 3 metabolites-13-00817-f003:**
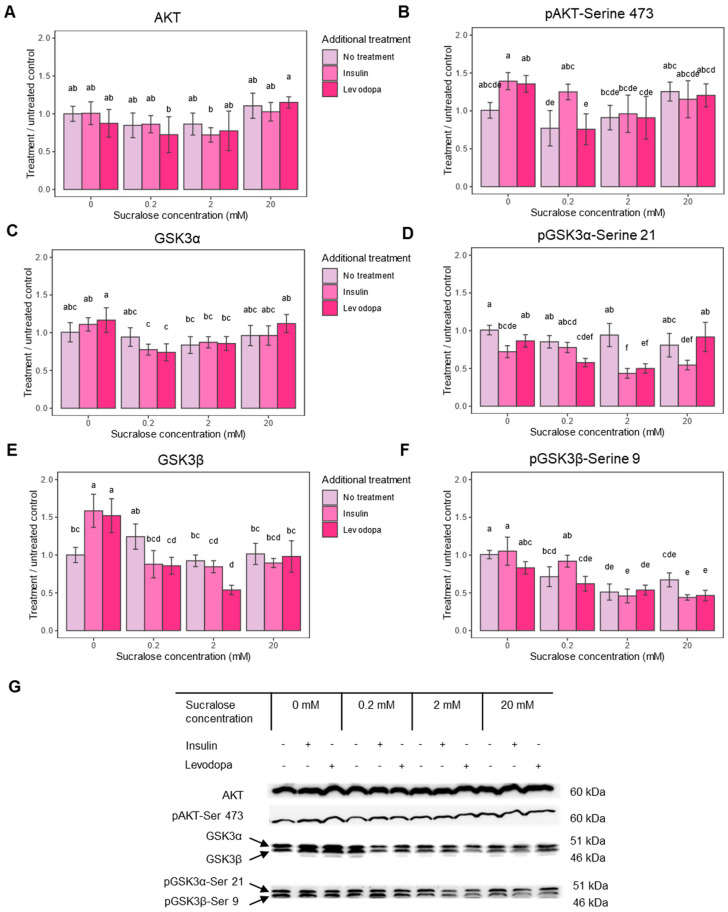
Insulin pathway epitope expression after treatment with sucralose, with and without an additional treatment with insulin or levodopa. Three treatment concentrations of sucralose were used (0.2 mM, 2 mM and 20 mM). **A** = protein kinase B (AKT), **B** = protein kinase B phosphorylated at Serine 473 (pAKT-Ser 473), **C** = glycogen synthase kinase 3 alpha subunit (GSK3α), **D** = glycogen synthase kinase 3 alpha subunit phosphorylated at serine 9 (pGSK3α-Ser 9), **E** = glycogen synthase kinase 3 beta subunit (GSK3β), **F** = glycogen synthase kinase 3 beta subunit phosphorylated at serine 21 pGSK3β-Ser 21, **G** = representative Western blots of respective epitopes. Epitope levels were calculated from signal intensity and presented as ratio of signal intensity to 0 mM sucralose/No additional treatment group. Two-way ANOVA (AKT), F(6,36) = 0.771, *p* = 0.5979; Two-way ANOVA (pAKT), F(6,39) = 2.050, *p* = 0.0819; Two-way ANOVA (GSK3α), F(6,37) = 2.487, *p* = 0.0402; Two-way ANOVA (pGSK3α-Ser21), F(6,36) = 8.721, *p* < 0.0001; Two-way ANOVA (GSK3β), F(6,36) = 11.374, *p* < 0.0001; Two-way ANOVA (pGSK3β-Ser 9), F(6,36) = 4.808, *p* = 0.0010; post HOC test Tukey HSD with *p*-value set at *p* < 0.05. Groups bearing the same letter above the graph are not statistically different.

**Figure 4 metabolites-13-00817-f004:**
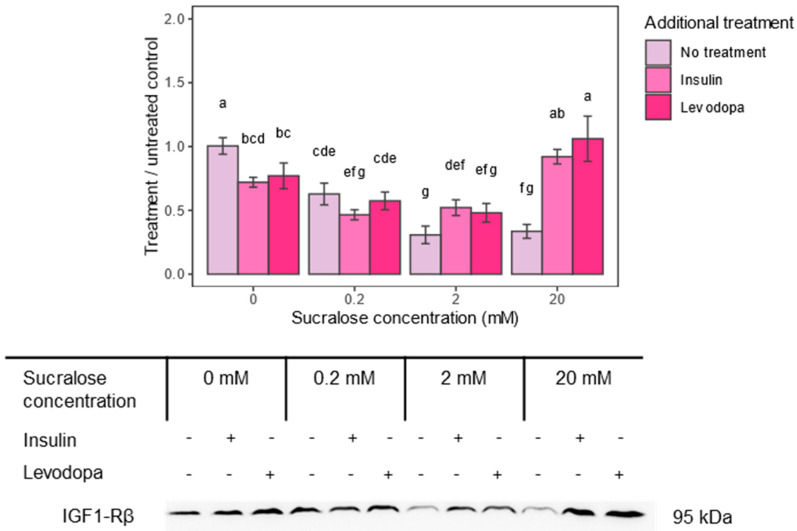
Insulin-like growth factor 1 receptor–beta subunit (IGF1-Rβ) epitope expression after treatment with sucralose and additional insulin or levodopa treatment. Three concentrations of sucralose were used (0.2 mM, 2 mM and 20 mM). Epitope levels were calculated from signal intensity and presented as ratio of signal intensity to 0 mM sucralose/No additional treatment group. Two-way ANOVA sucralose treated groups, F(6,36) = 33.171, *p* < 0.0001; post HOC test Tukey HSD with *p*-value set at *p* < 0.05. Groups bearing the same letter above the graph are not statistically different.

**Table 1 metabolites-13-00817-t001:** List of solutions and their composition.

Solution Name	Solution Composition
Phosphate buffered saline with protease inhibitors	1x PBS, 0.32 M sucrose (Acros Organics, Morris Plains, NJ, USA), 1 mM PMSF (Acros Organics, Morris Plains, NJ, USA), 5 mM NaF (Acros Organics, Morris Plains, NJ, USA), 1 mM Na_3_VO_4_ (Sigma-Aldrich, St. Louis, MO, USA), 1 mM EDTA (Sigma-Aldrich, St. Louis, MO, USA), 1 tablet of Complete Mini Protease Inhibitor on 10 mL of buffer (Roche, Basel, Switzerland)
MTT Solubilization Solution	10% Triton X-100 (Fisher Scientific, Fair Lawn, NJ, USA), 0.1 N HCl (Gram mol, Zagreb, Croatia) in anhydrous isopropanol (Gram mol, Zagreb, Croatia)
6x Laemmli buffer	0.35 M Tris base (pH 6.8) (Fisher Scientific, Waltham, MA, USA), 10% SDS (Sigma-Aldrich, Saint Louis, MO, USA), 30% glycerol (Acros Organics, Morris Plains, NJ, USA), 9.3% DTT (Acros Organics, NJ, USA)
Electrophoresis buffer	25 mM Tris base, 192 mM glycine (Fisher Scientific, Waltham, MA, USA) and 0.1% SDS
Towbin buffer	25 mM Tris base, 192 mM glycine, 20% methanol (Gram-Mol, Zagreb Croatia) in distilled water
1x phosphate buffer (PBS)	137 mM NaCl (Gram-Mol, Zagreb Croatia), 2.7 mM KCl (Kemika, Zagreb Croatia), 10 mM Na_2_HPO_4_ (Acros Organics, Morris Plains, NJ, USA), 1.8 mM KH_2_PO_4_ (Fisher Scientific, Loughborough, UK)
Bovine serum albumin (BSA) solution	3% BSA (Sigma-Aldrich, St. Louis, MO, USA) in 1x PBS-T

**Table 2 metabolites-13-00817-t002:** List of primary and secondary antibodies.

Antibody	Class	Origin	Manufacturer and Catalog Number	Catalog Number	Dilution Ratio
Anti-protein kinase B (AKT)	IgG, monoclonal	Mouse	Cell Signaling, Danvers, MA, USA	2920S	1:1000
Anti-protein kinase B–phosphorylated on serine 473 (pAKT)	IgG, monoclonal	Rabbit	Cell Signaling, Danvers, MA, USA	9271S	1:1000
Anti-glycogen synthase kinase 3 α + β (GSK3 α/β)	IgG, monoclonal	Rabbit	Cell Signaling, Danvers, MA, USA	5676S	1:1000
Anti-glycogen synthase kinase 3 alpha/beta phosphorylated at serine 21/serine 9 (pGSK3 α/β)	IgG, monoclonal	Rabbit	Cell Signaling, Danvers, MA, USA	9331S	1:1000
Anti-insulin-like growth factor receptor–β subunit (IGF1Rβ)	IgG, polyclonal	Rabbit	Santa Cruz, CA, USA	sc-713	1:500
Anti-mouse antibody labeled with biotin (αMO-biotin)	IgG	Goat	Jackson ImmunoResearch, West Grove, PA, USA	115–065-071	1:20,000
Anti-rabbit antibody labeled with biotin (αRB-biotin)	IgG	Goat	Jackson ImmunoResearch, West Grove, PA, USA	111–065-144	1:20,000

**Table 3 metabolites-13-00817-t003:** MALDI-TOF-MS analysis of cell metabolic profile after treatment with sucralose, sucralose + insulin, or sucralose + levodopa treatment.

Tentatively Annotated Compounds	KEGG ID	Adduct	*m*/*z*	Sucralose 0 mM			Sucralose 0.2 mM			Sucralose 2 mM			Sucralose 20 mM		
				No Treat.	Insulin	L-dopa	No Treat.	Insulin	L-Dopa	No Treat.	Insulin	L-dopa	No Treat.	Insulin	L-Dopa
Palmitoleic acid	C08362	M-H_2_O-H	235.2	**173,075** ± 1049						**51,885** ± 14,980		**31,168** ± 12,818			
Myristic acid	C06424	M+NH_4_	246.2	**0**					**3269** ± 844						
Arachidic acid	C06425	M-H	311.3	**109,061** ± 20,504			**53,144** ± 5819				**195,335** ± 35,076	**210,885** ± 45,574			**272,520** ± 55,464
Docosanoic acid	C08281	M+H-H_2_O	323.3	**11,556** ± 2554		**3889** ± 2740						**21,899** ± 1121			**5188** ± 2100
Oleic acid	C00712	M+HAc-H	341.3	**46,931** ± 8604									**0**		
Eicosenoic acid	C16526	M+HAc-H	369.3	**20,669** ± 2962				**44,696** ± 6296			**40,722** ± 8668				
4-Hydroxysphinganine	C12145	M+HAc-H	376.3	**0**										**68,161** ± 25,986	
4-Hydroxy-phenyl-acetyl-glutamic acid	C05595	M+Cl	316.1	**0**			**65,001** ± 22,413								
6-Hydroxymelatonin	C05643	M+NH_4_	266.2	**2528** ± 2416										**10,080** ± 1726	

Both polar and non-polar phases were analyzed in positive and negative mode using spot analysis. Obtained data were analyzed in R statistical software (v 4.0.3.) and compared against untreated control group using False Discovery Rate-corrected Welch *t*-test for significant *m*/*z* selection. During KEGG database search, only unique hits were retained, while multiple hits, including isobars, were excluded. *p*-value was set at *p* < 0.05, and only statistically significant data are shown. Data are presented as average signal intensity ± standard deviation. If the table cell has a missing number, it indicates values statistically similar to the untreated control group, and zero value indicates absence of metabolite in sample; N biological replica = 3; *m*/*z* = mass/charge ratio; No. treat. = No additional treatment; L-dopa = levodopa.

## Data Availability

The data presented in this study are available on request from the corresponding authors. The data are not publicly available due to lack of resources for data hosting.

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
