# Peer review of "Sucralose Targets the Insulin Signaling Pathway in the SH-SY5Y Neuroblastoma Cell Line"

_metabolites, 2023, doi:10.3390/metabo13070817_

Round 1
Reviewer 1 Report
In the present manuscript (Metabolites-2473212), Covic et al., have attempted to study the role of Sucralose a widely used NNS in modulating cellular signal transduction cascades in a dopaminergic cell-line and have tried to correlate with prevalent pathologies. This is an interesting topic given the wide-spread use of NNS and prevalence of diabetes, obesity and metabolic syndrome. However, the present study has several pitfalls.
Major comments:
1. Authors mention that Sucralose at a concentration of 2mM is closest to the permitted intake threshold…then why authors chose 20mM. It would be better to use concentrations below and not 10 times more.
2. Authors do not mention for how long the Sucralose treatment was done? It will be better if they present data’s of different duration of Sucralose treatment. Will the interpretations be different if the cells were maintained in low glucose medium, instead of high?
3. Along with MTT, PI-FACS and AnnexinV staining should be performed to know the effects on cell cycle and another measure of cell death due to various treatment. This is crucial as PI3K-AKT signaling is a major regulator of both cell proliferation and apoptosis.
4. What is the mechanism underlying Sucralose-induced reduction in pAKT levels?
5. The WB in Fig. 3 needs to be redone. They should be presented together one-below another for better comparisons. Nothing could be deciphered from the presented western blots of AKT and pAkt? The samples seem to have mixed between wells…giving a continuous band. All the quantifications could be provided separately. The authors have provided ‘raw data’ for their WBs. But, which one of them is used for the main Figure? Or are they different exposures. The raw data should show the labelled Mol. WT markers. For total AKT why only chemiluminescence raw data was shown?
6. It will be good to know the Akt kinase activity in different treated conditions.
7. In table 2, authors mention of using phospho-Tyrosine GSK3 antibodies, while in Fig.2 they present WBs with an entirely different phospho-GSK3 antibody (Ser9/Ser21). What was exactly measured and why this discrepancy?
8. In Fig. 4: authors should provide some control. Otherwise it is not possible to draw any conclusion. Is the reduction of IGF1R-B related to cell death due to Sucralose treatment?
9. The change in metabolite profile is induced through which receptor/signaling hub? Are they related to modulation of AKT and/or IGF1R pathways or independent of them?
Minor comment:
The text has a few spelling and grammatical errors which need to be corrected.
Author Response
Major comments:
- Authors mention that Sucralose at a concentration of 2mM is closest to the permitted intake threshold…then why authors chose 20mM. It would be better to use concentrations below and not 10 times more.
Thank you for your comment. In Materials and methods, we have added the reason for the selection of tested concentrations. Our choice was motivated by the finding of a tenfold difference in the concentrations of low-calorie sweeteners found in subjects who did not report their use before the study and were advised not to take them during the study, yet were exposed to them.
Sylvetsky AC, Walter PJ, Garraffo HM, Robien K, Rother KI. Widespread sucralose exposure in a randomized clinical trial in healthy young adults. Am J Clin Nutr. 2017 Apr;105(4):820-823. doi: 10.3945/ajcn.116.144402. Epub 2017 Feb 22. PMID: 28228424; PMCID: PMC5366047. (https://www.ncbi.nlm.nih.gov/pmc/articles/PMC5366047/)
Also, we used the highest dose in order to test the neurotoxicity. In other papers, the same or even higher concentrations (50mM and even 125 mM) were used on cell culture models.
Masubuchi Y, Nakagawa Y, Ma J, Sasaki T, Kitamura T, et al. (2013) A Novel Regulatory Function of Sweet Taste-Sensing Receptor in Adipogenic Differentiation of 3T3-L1 Cells. PLOS ONE 8(1): e54500. https://doi.org/10.1371/journal.pone.0054500
Bórquez JC, Hidalgo M, Rodríguez JM, Montaña A, Porras O, Troncoso R, Bravo-Sagua R. Sucralose Stimulates Mitochondrial Bioenergetics in Caco-2 Cells. Front Nutr. 2021 Jan 18;7:585484. doi: 10.3389/fnut.2020.585484. PMID: 33537337; PMCID: PMC7848014.
Omran A, Ahearn G, Bowers D, Swenson J, Coughlin C. Metabolic effects of sucralose on environmental bacteria. J Toxicol. 2013;2013:372986. doi: 10.1155/2013/372986. Epub 2013 Dec 3. PMID: 24368913; PMCID: PMC3866777.
- Authors do not mention for how long the Sucralose treatment was done? It will be better if they present data of different duration of Sucralose treatment. Will the interpretations be different if the cells were maintained in low glucose medium, instead of high?
Thank you for your comment. The duration of sucralose treatment is described in the first sentence of chapter 2.2 Treatments (Materials and Methods). We limited the treatment to 24 hours based on a previous pharmacokinetic study in which it was found that the effective half-life for sucralose is 13 hours, while the mean residence time is 18.8 hours.
Roberts A, Renwick AG, Sims J, Snodin DJ. Sucralose metabolism and pharmacokinetics in man. Food Chem Toxicol. 2000;38 Suppl 2:S31-41. doi: 10.1016/s0278-6915(00)00026-0. PMID: 10882816.
Given that the elimination of a dose of 1 mg/kg through urine and feces takes 5 days, we considered that it was sufficient to test only one period of time on cell cultures. An explanation has been added to chapter 2.2 Treatments (Materials and Methods).
We cannot say for sure how the cells would behave if they were exposed to sucralose in a low-glucose medium, but we assume that in that case insulin treatment would lead to an even greater inhibition of insulin signaling due to glucose-deprivation.
- Along with MTT, PI-FACS and AnnexinV staining should be performed to know the effects on cell cycle and another measure of cell death due to various treatment. This is crucial as PI3K-AKT signaling is a major regulator of both cell proliferation and apoptosis.
Thanks for the suggestion. We decided on the MTT viability test because it is based on the activity of mitochondrial enzymes. Given that the permitted dose of sucralose for humans is a very high 5 mg/kg, we believed that none of the doses used would lead to cell apoptosis - which turned out to be correct. On the other hand, insulin resistance can be considered an acquired mitochondrial disease, so the result of the MTT test is interesting precisely in this sense. The highest concentration of sucralose had a positive effect on survival, which is in favor of tumorigenic transformation of cells and their generally better survival. Our focus was not on the investigation of the mechanism of tumorigenesis, but we agree that it is an interesting topic. In order to do that, research should be repeated on several different cell lines, and such works already exist.
- What is the mechanism underlying Sucralose-induced reduction in pAKT levels?
Thank you for the stimulating question. Based on our results, we can assume that the reduction in pAKT level is related to down-regulation of insulin-like growth factor 1 signaling. Another option is that excessive stimulation of sweet taste receptors led to the activation of negative feedback loops, but we have no evidence for this hypothesis.
Zheng WH, Quirion R. Insulin-like growth factor-1 (IGF-1) induces the activation/phosphorylation of Akt kinase and cAMP response element-binding protein (CREB) by activating different signaling pathways in PC12 cells. BMC Neurosci. 2006 Jun 22;7:51. doi: 10.1186/1471-2202-7-51. PMID: 16792806; PMCID: PMC1534052.
An explanation has been added to the discussion.
- The WB in Fig. 3 needs to be redone. They should be presented together one-below another for better comparisons. Nothing could be deciphered from the presented western blots of AKT and pAkt? The samples seem to have mixed between wells…giving a continuous band. All the quantifications could be provided separately. The authors have provided ‘raw data’ for their WBs. But, which one of them is used for the main Figure? Or are they different exposures. The raw data should show the labelled Mol. WT markers. For total AKT why only chemiluminescence raw data was shown?
Thank you for requesting this improvement to our manuscript. Figure 3 was redone as requested. We identify specific masses on the chemiluminescence raw images, and mark which of these was used for the image in the paper (in Folder named Membranes labelled with marker).
Chemiluminescence images and raw data were not obtained from different exposures of the same membrane, but from different membranes. The experiment was performed in minimum triplicate (for example, pAKT was repeated 6 times and all experiments are included in final quantification). The gel presented in the paper is archived in supplementary data (named as Used in main image). In our experiments, epiluminescent photographs are captured in the event of the emergence of multiple unexpected bands. These epiluminescent images are primarily employed during postprocessing, to accurately ascertain appropriate signals. Only chemiluminiscence data was used for quantification. Our Supplementary Data Section contains results from all experiments performed in our study. Quantification for each membrane could be provided upon request to the corresponding authors of our paper.
- It will be good to know the Akt kinase activity in different treated conditions.
Thank you for your comment. In the answer to question 1, we explained the reasoning behind the selected concentrations of sucralose. In the answer to question 2, we explained why only one treatment time interval and high-glucose medium was used. In the same answer, the result that could be expected in a low-glucose medium was explained. We put our effort into understanding a total of 9 different treatment conditions (3 concentrations of sucralose x 3 treatments - no treatment, insulin, levodopa).
- In table 2, authors mention of using phospho-Tyrosine GSK3 antibodies, while in Fig. 2 they present WBs with an entirely different phospho-GSK3 antibody (Ser9/Ser21). What was exactly measured and why this discrepancy?
We apologize for the mistake and thank you for noticing it. In this study, only antibodies against phospho-Ser epitopes were used. The error occurred because the materials and methods were written by a collaborator who did not participate in the western blot analysis. The error has been corrected in Table 2.
- In Fig. 4: authors should provide some control. Otherwise it is not possible to draw any conclusion. Is the reduction of IGF1R-B related to cell death due to Sucralose treatment?
Thank you for your question. In Figure 4, the controls were cells that were not treated with sucralose (sucralose concentration 0 mM) and cells that were treated only with insulin or levodopa, which were compared with cells pre-treated with 3 concentrations of sucralose (0.2 mM, 2 mM, 20 mM), and insulin or levodopa. Given that sucralose at a concentration of 0.2 mM and sucralose at a concentration of 20 mM did not lead to a significant decrease in cell viability, we believe that cell death was not the prime reason for the down-regulation of IGF1-Rβ.
- The change in metabolite profile is induced through which receptor/signaling hub? Are they related to modulation of AKT and/or IGF1R pathways or independent of them?
Thank you for the stimulating question. Given that part of the changes in the metabolic profile of the cell occur already upon stimulation with sucralose (at 0.2 and 2 mM concentration), it can be assumed that they are triggered by sweet taste receptors or cross-talk between sweet taste receptor and IGF1 signalling pathway, in this case by its deprivation. Activation of sweet taste receptors is associated with production of the second messenger cAMP, activation of protein kinase A and phosphorylation of CREB (cAMP response element-binding protein). Genes related to lipid synthesis, such as fatty acid synthase (FAS), have CRE (cAMP response element) in their promotor sequence. On the other hand, IGF1 acts on the PI3K/AKT pathway, which also leads to an increase (or reduction in the case of deprivation) of FAS and ACC (acetyl-CoA carboxylase expression). In this case, due to the reduction of pGSK3, there will be less phosphorylation of SREBP-1c (Sterol Regulatory Element-Binding Protein-1c), its stabilization will occur and lipid synthesis will be stimulated. Both described mechanisms contribute to insulin resistance.
Minor comment:
The text has a few spelling and grammatical errors which need to be corrected.
Thank you for your comment. The text has undergone English editing.

Reviewer 2 Report
This paper studied the impact of non-nutritional sweeteners on neurogenesis.SH-SY5Y cell line differentiated from Parkinson's disease (PD) .Cells were exposed to sucralose alone or in combination with insulin or levodopa.To verify the metabolic effect of Sucralose on the insulin signaling pathway in SH-SY5Y cell line cell model, the activation of the insulin signaling pathway was evaluated by detecting Protein kinase B (AKT), glycogen synthase kinase 3 (GSK3) and their phosphorylation forms or Insulin-like growth factor 1 receptor (IGF1-R) using Western blot. The metabolic effects of different treatments were measured using MALDI-TOF MS analysis. Finally, this paper shows that Sucralose interferes with the metabolism of differentiated SH-SY5Y cell lines, and the metabolic response is mainly determined by the changes in lipid composition and has a potential impact on the insulin signaling pathway. This paper has positive implications for the use of non-nutritional sweeteners in the field of neurogenesis.
There are some issues that must be solved before it is considered for publication.
1. The Introduction section does not provide a sufficient introduction to the research background. For example, the authors should add some additional information about Levodopa.
2. Only Western blot was used to verify protein expression, and the experimental method was too simplistic. Other methods can be added to verify the expression of the target protein and increase credibility.
3. Table 3 is somewhat blurry and affects the appearance. Please replace it with a clearer table for easy reading.
Author Response
Response to reviewer 2
- The Introduction section does not provide a sufficient introduction to the research background. For example, the authors should add some additional information about Levodopa.
Thank you for your comment. We have expanded the introduction with additional information about insulin and levodopa, which were used as treatment in our study. We also noticed that we did not add the main results related to the use of insulin and levodopa in the abstract, so that was also added.
- Only Western blot was used to verify protein expression, and the experimental method was too simplistic. Other methods can be added to verify the expression of the target protein and increase credibility.
Thanks for the comment. We completely agree. In such a short time (10 days) we are not able to add experiments because only the differentiation of neurons lasts 3 weeks. Despite this, we think that the paper, even in this form, is solid evidence of insulin signalling disruption. Also, our results complement each other, so this also speaks in favour of the accuracy of the findings.
- Table 3 is somewhat blurry and affects the appearance. Please replace it with a clearer table for easy reading.
Thank you for noticing. The table has been replaced with a more readable version.

Round 2
Reviewer 1 Report
The revised manuscript has sufficiently improved and could be accepted in the present form.
Fine.
Reviewer 2 Report
Accept